# Quantifying Entropy in Response Times (RT) Distributions Using the Cumulative Residual Entropy (CRE) Function

**DOI:** 10.3390/e25081239

**Published:** 2023-08-21

**Authors:** Daniel Fitousi

**Affiliations:** Department of Psychology, Ariel University, Ariel 40700, Israel; danielfi@ariel.ac.il

**Keywords:** entropy, differential entropy, response times, survivor function, processing capacity

## Abstract

Response times (RT) distributions are routinely used by psychologists and neuroscientists in the assessment and modeling of human behavior and cognition. The statistical properties of RT distributions are valuable in uncovering unobservable psychological mechanisms. A potentially important statistical aspect of RT distributions is their entropy. However, to date, no valid measure of entropy on RT distributions has been developed, mainly because available extensions of discrete entropy measures to continuous distributions were fraught with problems and inconsistencies. The present work takes advantage of the *cumulative residual entropy* (CRE) function—a well-known differential entropy measure that can circumvent those problems. Applications of the CRE to RT distributions are presented along with concrete examples and simulations. In addition, a novel measure of instantaneous CRE is developed that captures the rate of entropy reduction (or information gain) from a stimulus as a function of processing time. Taken together, the new measures of entropy in RT distributions proposed here allow for stronger statistical inferences, as well as motivated theoretical interpretations of psychological constructs such as *mental effort* and *processing efficiency*.

## 1. Introduction

Claude Shannon’s seminal work [1,2] has exerted a considerable impact on many fields of science. One of those fields is psychology. Shannon’s precise definitions of such concepts as information, entropy, and channel-capacity revolutionized the way psychologists thought about human cognition and how they measured it [3,4,5,6,7,8]. The mind (and later the brain) was construed as a processing device—one that can encode, manipulate, and transmit information [9]. From a historic perspective, information theory served as a major catalyst in launching the cognitive revolution [10]. Consider, for example, Shannon’s notion of a limited-capacity channel. This idea was (and continues to be) instrumental in research on human attention and memory [11,12], as it was originally consequential for the discovery of capacity limitations on short-term memory [6] or for the uncovering of *Hick’s law* [13], a remarkable regularity by which mean response times (RT) increases linearly with the *log* number of choice response alternatives. The discovery of Hick’s law and other conceptual advances [14] have turned RT into one of the central methodologies in psychological theorizing and experimentation [15,16,17,18]. The main assumption guiding practitioners of RT methodologies is that observed RT reflect the duration of internal cognitive processes [19]. The more demanding a cognitive operation is, the longer its completion time [16]. Most researchers rely primarily on mean RT as their dependent variable. However, there is now an emerging consensus among psychologists that RT distributions conceal more information than meets the eye [20,21,22] and that this fine-grained information should be extracted to advance information processing models [23]. Consequently, recent years have seen a rise in the number of distributional RT measures and models for review; see [17]. These approaches aim at capturing broader statistical aspects of the RT distributions to provide a richer and more fine-grained data set against which human cognition can be modeled and tested. One of the most theoretically valuable quantities lurking in RT distributions is their entropy [24]. Entropy is closely related to such psychological constructs as *processing capacity* [11,25], *workload* [26], and *efficiency* [27], which are often measured using RT methodologies [28,29].

However, to date, no valid measure of entropy on continuous RT distributions has been developed. The reason for this is that available extensions of discrete entropy measures to continuous distributions (i.e., differential entropy) were fraught with problems and inconsistencies [30]. First, the entropy of discrete distributions is always positive, whereas the differential entropy of continuous distribution can take any value on the extended real line. Second, the entropy of empirical distributions cannot be used to approximate the differential entropy of theoretical continuous distributions. Third, Shannon’s entropy does not take into consideration the content represented by probabilities, but only their probabilistic values. This is unfortunate because the precise quantitative characterization of total and instantaneous entropy in RT distributions can provide considerable statistical and theoretical advantages. One notable benefit has to do with measurement. A valid entropy measure will allow comparison across various tasks, irrespective of those tasks’ units of measurement, by adhering to a single quantity of entropy (e.g., bits). Another contribution is in introducing richer theoretical accounts of abstract psychological constructs such as *processing capacity* [25] and the ability to associate these psychological constructs with their corresponding entropy signatures in the brain [31,32,33]. Fortunately, recent advances in information theory offer a new measure of differential entropy in continuous distributions. It has been developed by Rao and colleagues [30,34,35,36] and is known as the *cumulative residual entropy* (CRE). This measure successfully extends the Shannon entropy to random variables with continuous distributions thanks to three major properties: (a) it is consistently defined in both the continuous and discrete domains, (b) it is always nonnegative, and (c) it can be computed from empirical data and asymptotically converge to true values [30]. These properties of the CRE make it a viable measure of entropy in continuous distributions in many domains of application [36,37].

The present work takes advantage of these developments and imports the CRE to the field of psychology. Here, I demonstrate how differential entropy measures on empirical and theoretical RT distributions can be constructed. This approach allows for stronger statistical inferences and well-motivated theoretical interpretations of cognitive phenomena. The remainder of the paper is structured as follows. I start by introducing the canonical RT functions deployed in psychology. I then introduce the differential entropy measure—CRE [30,34,35]—and show how it can be applied to RT distributions. In addition, a novel measure of instantaneous cumulative residual entropy [iCRE(t)] is developed that captures the momentary entropy at time *t*, and its slope gauges the rate of information gain. Finally, the relations of these novel entropy measures to existing RT functions and psychological concepts such as *mental effort* and *processing capacity* are discussed.

## 2. Basic Tools for Stochastic Modeling of RT

Cognitive phenomena are not deterministic but are rather subjected to a certain degree of uncertainty and variability [16]. This idea was succinctly put by Ebbinghaus [38], who referred to “the constant flux and caprice of mental events” (p. 19). Psychologists widely agree that models of human behavior and cognition should incorporate stochastic mechanisms. To capture the stochastic nature of cognitive processes, RTs are treated as a *random variable* [16,18] and are represented by empirical and theoretical continuous distributions. In general, there are two primary practices for making inferences based on RT distributions. Proponents of the *parametric approach* fit a theoretical distribution to empirical RT data and use the estimated parameters in building and testing cognitive models [39]. For example, the *exGaussian* distribution [21,40,41,42] has often been used [43]. This distribution closely mimics many aspects of empirical RT distributions, such as their negatively skewed shape and long right tail. It also provides excellent fits to experimental data. I use this distribution throughout this paper in simulations and demonstrations. An RT in this distribution is the sum of a random value from a Gaussian distribution with mean μ and standard deviation σ and a random value from an exponential distribution with mean τ. The density function is
(1)$$\begin{array}{c} f\left(t\right)=\frac{e^{-[\left(t-\mu \right)/\tau -\sigma^{2}/\left(2\tau^{2}\right))]}}{\tau \sqrt{2\pi }}\int_{-\infty }^{\left[(t-\mu )/\sigma ]-\sigma /\tau \right)}e^{-y^{2}/2}dy \end{array}$$
with mean μ+τ and standard deviation $\sqrt{\left(\sigma^{2}+\tau^{2}\right)}$. Figure 1 presents the outcome of simulations with the exGaussian distribution, demonstrating how changes in parameters (μ, σ, and τ) induce changes in the location, scale, and shape of the distribution. The initial distribution parameters were set to μ=400, σ=30 and τ=60. In each simulation, one of the parameters was increased to demonstrate its influence; see [21,44].

An alternative practice of modeling RT distributions is the *aparametric approach*. Proponents of this practice make distributional-free inferences based on moments and cumulants [45,46].

In both the parametric and non-parametric approaches to RT distributions, researchers harness a number of probability-theory continuous functions to model stochastic aspects of information processing [16]. The first tool is the *probability distribution function* (PDF). The PDF (see Figure 2a) measured with RT has a single mode and is negatively skewed with a long tail extending to the right [43]. However, the stochastic nature of information processing is not immediately discernible from the PDF. To capture these aspects directly, researchers harness the *cumulative distribution function* (CDF), F(t)=Pr{T≤t}, where *t* indicates any value of the continuous random variable of time, and *T* stands for one specific value. Townsend and Ashby [16] noted that “... it may frequently turn out that the distribution function F(t) is much easier to calculate directly than is the density function f(t). Once the distribution function is known, the density function can easily be derived through Eq. 3.1 [which is the derivative of F(t)]” (p. 25). F(t) gives the probability that the participant has responded by time *t* (see Figure 2b). This function starts at 0 and levels off at 1, when there is certainty that the participant has responded. A cumulative probability of 0.9 at some time *t*, for example, means that 90% of responses have terminated by that time. The PDF and the CDF are related such that the CDF is the integrated PDF from 0 to the current *t*:(2)$$\begin{array}{cc} F(t)= & \int_{0}^{t}f\left(t^{\prime }\right)dt^{\prime } \end{array}$$
Reciprocally, the PDF at *t* is the derivative of the CDF $f\left(t\right)=dF\left(t^{\prime }\right)/dt^{\prime }$.

A third function that is often deployed in the RT literature is the *survivor function*, S(t), which captures the probability that the participant has not responded yet by time *t*, i.e., S(t)=Pr{T>t} (see Figure 2c). Note that S(t) and F(t) complement each other:(3)$$\begin{array}{cc} F(t)=Pr\{T\leq t\}= & 1-Pr\{T>t\}=1-S(t) \end{array}$$

Another function that plays an important role in RT modeling is the *hazard function* [47]. This function is well-established in longitudinal or distributional (duration, transition, or failure-time) analysis [48]. The survival technique has already been applied in many areas of psychology [47,49,50,51]. Townsend and Ashby [11] were the first to apply it to RT distributions:(4)$$\begin{array}{cc} h(t)= & \lim_{\Delta t\to 0}\frac{P(t\leq T\leq t+\Delta t|T\geq t)}{\Delta t} \end{array}$$(5)$$\begin{array}{cc} = & \frac{f(t)}{1-F(t)}=\frac{f(t)}{S(t)} \end{array}$$
where the numerator is the probability density function, f(t), for task completion times, giving the probability that the task will be completed at some particular time *t*, and the denominator is the survivor function, S(t), designating the probability that the task will not have been completed by time *t*. The hazard function is a conditional probability function, providing the likelihood that the process fails in the next instant, given that it has not yet failed (see Figure 2d). A decreasing hazard function means that as the process gets longer, the likelihood is higher that it will terminate in the next instant. An increasing hazard function entails that as the process continues, the likelihood is high that it will not terminate on the next instance. A constant hazard function suggests that the likelihood that the process will terminate on the next moment is independent of its duration to that moment [40]. The shape of the hazard function can be used to decide among classes of cognitive models, evaluate models, and generate predictions for RT distributions [18].

In the psychological modeling of response times, the hazard function h(t) has often been interpreted in terms of the *energy* or *efficiency* invested by a participant in a task [11,24,45,50,51,52]. The logic behind this is that the hazard function expresses the likelihood of a worker completing the task in the next instant, given that the worker has not yet completed the task. A worker with low capacity has a low likelihood of completing the task in the next instance given that she has not finished it yet, whereas a worker with high capacity has a high likelihood of completing the task in the next moment given that she has not completed it yet. Finally, I introduce the related *integrated hazard function H(t)* [16]:(6)$$\begin{array}{ccc} H(t) & = & \int_{0}^{t}h\left(t^{\prime }\right)dt^{\prime }=\int_{0}^{t}\frac{f(t^{\prime })}{S(t^{\prime })}=-\log S\left(t^{\prime }\right) \end{array}$$
which takes the integral of the hazard function from 0 to time *t*. The function starts at 0 and increases monotonically to infinity (see Figure 2e). The integrated hazard function also equals −log[S(t)]—a mathematical identity that greatly assists researchers in simplifying computations [11,12,53]. The value of the *cumulative hazard function* up to time *t* has been interpreted as the cumulative amount of work done or effort expanded in a task by time *t* [51].

The hazard (and the integrated hazard) function meets the desirable characteristics of any viable measure of dynamic capacity [53]: (a) it is bounded at the lower end by zero and bounded on the upper end by the physical limits of the system, (b) its integral captures the cumulative amount of work accomplished by some time, and (c) the cumulative hazard itself is bounded at the lower end by zero and unbounded at the upper end. These aspects are consistent with extant approaches to *capacity* in contemporary psychological literature [54]. These approaches often relate to the efficiency by which information is processed, bearing close affinity to the idea of “bandwidth” or energy invested per unit of time. As I show in the next sections, these beneficial statistical aspects also characterize the entropy measures developed here for RT distributions.

The five RT functions presented here are critical for the construction of our novel entropy measures. Figure 2 presents simulations of these five functions using the exact exGaussian parameters from the previous simulation.

It is important to note that capacity can be measured at different levels of granularity. It can be assessed at the level of the *system*, as is the case when participants process entire faces composed of individual features (e.g., nose, eyes). The capacity of a system can be potentially affected by the global workload (i.e., number of featural elements to be processed) [45,51]. The more elements to be processed, the greater the capacity needed to complete the processing. Alternatively, capacity can be assessed at the local level of the *individual elements* that compose a system [53] (e.g., processing the nose in a face).

Many psychologists treat the processing of tasks/stimuli/elements in terms of a set of (in)dependent channels, along which information accumulation takes place [16,26,54]. Processing along each channel is modeled with the canonical RT functions. Psychologists apply the term “channel” as a metaphor, without actually deploying the standard entropy metric. The metaphor likely attained traction in the heydays of the cognitive revolution and has remained fruitful in psychology ever since. For example, in James Townsend’s seminal work [12,45], *serial* or *parallel* processing systems are modeled as a set of corresponding channels, along which accumulation of evidence takes place either one-channel-after-the-other (i.e., serial architecture) or simultaneously across all channels (i.e., parallel architecture). The capacity of a single channel is measured by the *integrated hazard function*, and general formal derivations quantify the joint capacity of the system for *serial* or *parallel* channel architectures, contingent on the operative *stopping rule* adopted by the observer (i.e., *self-terminating* or *exhaustive*) [45,55].

## 3. The Cumulative Residual Entropy (CRE)

Shannon [1,56] proposed entropy as a measure of *uncertainty U* in discrete distributions: (7)$$\begin{array}{ccc} U(F) & = & -\Sigma p_{i}\log p_{i} \end{array}$$
where $p_{i}$s are the probabilities derived from the distribution *F*. This measure of uncertainty has important properties: (a) it is always positive, (b) it vanishes *if and only if* it is a certain event, (c) entropy is increased by adding an independent component, and (d) entropy is decreased by conditioning. One may think of the following differential entropy as a possible extension: (8)$$\begin{array}{ccc} U(F) & = & -\int f(x)\log f(x)dx \end{array}$$

However, as noted at the outset, this measure leads to various challenges and inconsistencies. First, whereas the entropy of discrete distributions is positive, this differential entropy of continuous distributions can also be negative. Second, empirical distributions cannot be used to approximate this differential entropy of theoretical continuous distributions. Third, in Shannon entropy, the content represented is not taken into consideration, only the probabilistic values. Fourth, it is only defined for distributions with densities. Fifth, in contrast to discrete entropy, the reduction in the conditional differential entropy of *X* given *Y* does not imply that is *X* a function of *Y*. To solve all these problems, Rao and colleagues [30,34] presented the *cumulative residual entropy* (CRE) of *X*. The CRE extends the Shannon entropy to random variables with continuous distributions, and ameliorates all of the noted problems: (9)$$\begin{array}{ccc} \epsilon \left(x\right) & = & -\int S\left(x^{\prime }\right)\log S(x^{\prime })dx^{\prime } \end{array}$$
where S(x) is the survivor function, which captures the probability that an event has not yet occurred for values of X>x. The CRE has several marked properties: (1) CRE is consistently defined in both the continuous and discrete domains; (2) CRE is always nonnegative; (3) CRE can be computed from empirical data, and these computations asymptotically converge to the true values; and lastly, (4) the conditional CRE of *X* given *Y* is zero *if and only if X* is a function of *Y*. All of these properties are supported by the rigorous proofs given in [30]. These authors also proved that the CRE of the empirical distribution almost surely converges to the true CRE. Their proof also provides a way of computing the CRE for empirical distributions.

Equipped with knowledge of basic RT functions and the differential entropy measure CRE, we are now ready to move on to quantifying the entropy of RT distributions.

## 4. Application of the CRE to RT Distributions

The application of the CRE to RT distributions is straightforward. We express the cumulative residual entropy (CRE) in terms of the RT survivor function [30]:(10)$$\begin{array}{ccc} \epsilon \left(t\right) & = & -\int S\left(t^{\prime }\right)\log S(t^{\prime })dt^{\prime } \end{array}$$
which gives the amount of entropy in the RT distribution within the required bounds of integration. The integral of the CRE can be computed either analytically, for well-defined theoretical distributions (e.g., exponential distribution), or numerically, for empirical distributions, using dedicated methods such as linear or natural spline interpolation (as shown later). In the latter case, the computation of the entropy measure does not require the researcher to assume any specific parametric distribution. One can use only the corresponding survivor function. This is a marked advantage over parametric approaches to RT distributions that fit a theoretical distribution (e.g., exGaussian) and use the best-fitting parameters’ values in making inferences; see for example [57]. The CRE is measured in units of information. When the base of the logarithm is 2 (i.e., $log_{2}$), the units of information are called “bits”; when the base of the logarithm is 10 (i.e., $log_{10}$), the units of measurement are dubbed “bans”. One ban equals approximately $log_{2}10\left(\approx 3.322\right)$ bits.

I start by providing an example of an analytic solution of the CRE. The exponential distribution is a well-known theoretical distribution in the RT literature. It has been used extensively in the stochastic modeling of cognitive phenomena due to its memory-less properties [16]. An exponential distribution with mean $\frac{1}{\lambda }$ has the following PDF: (11)$$\begin{array}{c} p\left(t\right)=\left\{\begin{array}{cc} \lambda e^{-\lambda t} & \;t\geq 0\;\\ 0 & t<0\; \end{array}\right. \end{array}$$

The CRE of the exponential distribution amounts to
(12)$$\begin{array}{ccc} \epsilon \left(t\right) & = & -\int_{0}^{\infty }e^{-\lambda t^{\prime }}\log e^{-\lambda t^{\prime }}dt^{\prime }\\ & = & \int_{0}^{\infty }\lambda t^{\prime }e^{-\lambda t^{\prime }}dt^{\prime }\\ & = & \frac{1}{\lambda } \end{array}$$
which entails that the total amount of entropy of the exponential distribution equals its mean in information units. Thus, it is the case that exponential distributions with higher means also have larger amounts of total entropy.

## 5. Simulations of the CRE

Next, I simulated data from parametric distributions [40] (the exGaussian and exponential). The goal of these simulations was twofold: (a) to demonstrate the CREs of different distributions, and (b) to provide a practical example on how to calculate the integral of the CRE using non-analytic methods afforded by the statistical environment R [58].

Figure 3a–i depicts simulations with the exGaussian distribution. Two hypothetical experimental tasks were assumed. In three simulations, I have independently manipulated each of the three exGaussian parameters (μ, σ, and τ) across the two tasks (distributions). In the first simulation, I have manipulated the μ parameter across the two distributions but kept the σ and τ identical. One of the distributions (blue) was given a μ value of 800 compared to the baseline distribution (pink, μ=400). Recall that in the exGaussian distribution, the mean and the variance are affected differently by the manipulation of the parameters. The mean is composed of the μ and τ parameters [E(x)=μ+τ], whereas the variance is composed of the σ and τ parameters [$Var\left(x\right)=\sigma^{2}+\tau^{2}$]. Therefore, changes in μ selectively influence the mean of the distribution. As can be noted in Figure 3a, the PDF of the fast task illustrated in pink is shifted to the left of the slow task (illustrated in blue), but the shapes of the two distributions are identical. This shift in location can also be noted in Figure 3b, where the survivor function of the fast task (pink) is located to the left of the slow task (blue). This entails that the probability that the fast task will have been completed by time *t* is higher than that of the slow task. Most importantly, Figure 3c depicts the CRE functions [−S(t)logS(t)] for the two tasks. The CREs are similar, but again, the CRE of the easier task is shifted to the left. To calculate the entropy of these CREs, I have deployed the *AUC* (area under the curve) tool, which incorporates a numerical integration algorithm and is available as part of the *MESS* R package [59]. AUC uses linear or natural spline interpolation for two vectors to compute the area under the curve. In our case, the first vector contains the *t* values, while the second vector contains the corresponding [−S(t)logS(t)] values. The integration was held from 0 to *∞* (a practical value of 2500 ms was used as the upper limit of the integral). As a quick visual inspection of Figure 3c may suggest, the cumulative residual entropy of the two distributions was found to be comparable, amounting to approximately 66 bans.

In a second simulation, I manipulated the σ parameter across the two distributions but kept the μ and τ identical. One of the distributions (blue) was given a σ value of 100 compared to the baseline distribution (pink, σ=30). This manipulation selectively influences the variance of the distribution. The results of these simulations can be seen in Figure 3d–f. The blue PDF distribution has higher variance than the pink distribution, which is also reflected in the corresponding CRE functions. Using the AUC integration tool, I found that the distribution with the higher variance also produced larger total cumulative residual entropy (108 bans) compared to the distribution with the lower variance (66 bans).

In a third simulation, I set the τ parameter in the blue distribution to 120 (compared to the baseline pink distribution for which τ=60). This manipulation affects both the mean and the variance of the distribution because E(x)=μ+τ and $Var\left(x\right)=\sigma^{2}+\tau^{2}$. The outcome of this simulation can be seen in Figure 3g–i. This manipulation, which doubled the value of τ, resulted in higher variance in the PDF and larger CRE in the blue distribution compared to the pink distribution (151 bans compared to 66 bans).

Another set of simulations was held with the exponential distribution. As can be noted in Figure 4a, the task depicted by the green color is slower than the one illustrated in red (it has a lower λ rate). The difference in values of the λ parameter also entails that their corresponding PDF, survivor, and CRE functions differ in shape (see Figure 4b,c). Visual inspection of Figure 4c suggests that the areas under the two corresponding CRE functions are not equal. Indeed, the computation of the integral of the CREs using the AUC tool from 0 to *∞* (a practical value of 30 was used was used as the upper limit of the integral) revealed that the slow task yielded higher cumulative residual entropy (=1.96 bans) than the fast task (=1.07 bans). These values are in accordance with the analytic solution (see Equation (13)). The λ for the slow task amounts to $\frac{1}{1.96}=0.5$, while the λ for the fast task $\frac{1}{1.07}=0.9$. These are the exact parameter values initially used in the simulation.

The cumulative residual entropy (CRE) of RT distribution provides a viable measure of the amount of entropy induced by a task. From a psychological point of view, the CRE can be interpreted as reflecting the total amount of *energy* [11] or *effort* [25] invested by the participant in accomplishing the task at hand. The more capacity a task demands, the larger the amount of entropy it produces. This interpretation is consistent with extant psychological models of processing capacity and attention [45,46].

One notable aspect of the simulations is the possibility that mean RT and entropy, as measured by the CRE, are two independent aspects of RT distributions. The simulations showed that speed of processing and entropy are not necessarily dependent on each other. We could simulate RT distributions that have different means but identical CREs; conversely, we could generate RT distributions with the same mean RT but different CREs. However, in psychological research, a task that produces, on average, shorter mean RT is considered to be easier to perform, and is also expected to yield lower variance [60]. Because our conclusions are based only on simulated data, an open question still remains to be answered of whether a fast task/condition is also characterized by lower values of entropy than a slow task/condition. This hypothesis can be tested empirically with data from real experiments.

Another related point that deserves a comment concerns the relations between variance and entropy. It is likely that RT distributions with higher variance also have a larger amount of CRE. After all, variance is another measure of uncertainty [61,62]. This intuition was not refuted by the current simulations. But the entropy metric is preferable to the statistical measure of variance due to its richer theoretical import.

## 6. Instantaneous CRE

The application of the CRE to the RT distribution can give us the total amount of entropy in an experimental condition/task. This is very useful when comparing performance across conditions/tasks. However, in the stochastic modeling of response times [11], psychologists are also interested in *instantaneous* aspects of processing. One notable example for the relations between global and instantaneous measures of processing is the *hazard function* and *integrated hazard function*. The former is the derivative of the latter, giving the amount of energy or effort invested in a task at a given moment in time. In a similar vein, we can use the CRE to gauge the instantaneous entropy at a given point in time *t*. This can be accomplished by taking the derivative of the CRE at time *t*:(13)$$\begin{array}{c} \begin{array}{c} \epsilon^{\prime }\left(t\right)=\frac{d\epsilon (t)}{dt}=-S\left(t\right)\log S(t) \end{array} \end{array}$$

Integrating from t=0 to t=∞ will give us back the CRE. However, this quantity is a weighted measure because the entropy of each event [−log(x)] is weighted by its probability of occurrence *x* [56]. Thus, the unweighted cumulative residual entropy at time *t* equals
(14)$$\begin{array}{ccc} iCRE(t)= & = & -\log[S(t)] \end{array}$$
where iCRE(t) is a function that quantifies the instantaneous cumulative residual entropy at time *t*. It is a positive, linearly increasing function bounded between 0 and *∞* (see Figure 5b or Figure 5d). This function meets the desirable mathematical aspects of a capacity measure assessed across the course of a processing event. It is bounded at the lower end by zero and is unbounded at the upper end [50,53]. Figure 5b presents the iCRE(t) for two exGaussian distributions. As can be noted, values of the iCRE(t) increase linearly from zero to infinity as a function of time. The function is related directly to the survivor function and can be interpreted as the amount of information gained from the stimulus or alternatively the reduction in uncertainty with time. The iCRE(t) gives the amount of information gained at time *t* and is measured in units of information (e.g., bits). When the processing of a stimulus has not been started yet, the probability that processing has not been finished yet is 1, so the amount of information gained from this stimulus at this moment in time is 0. However, as S(t) decreases from 1 toward 0, more and more information is gained. This assumption about the accumulation of information from the stimulus is a cornerstone of *RT diffusion* models [63]. Interpreting the iCRE(t) as a reduction in entropy is also analogous to saying that the certainty about the stimulus/task’s final state of processing is increasing. Thus, when S(t) is equal 0, iCRE(t)=∞, entailing that the information one gains with respect to the hypothesis that processing has finished by time *t* is very large. This function can also stand for the self-information of a processing channel [56]. When it reaches infinity, or practically a relatively high value, one can safely conclude that processing has ended.

The iCRE(t) inherits the properties of the CRE: (1) it is consistently defined in both the continuous and discrete domains, (2) it is nonnegative, (3) it can be computed from empirical data, and (4) it increases by the addition of another variable (processing channel) *if and only if* the variables (channels) are independent. Another property of the iCRE(t) is that it is increasing monotonically as a function of processing time. All of these make it an excellent measure of processing capacity.

## 7. The Slope of the iCRE(t) as a Measure of Information-Gain Rate

The slope of the iCRE(t) can provide us with another valuable aspect of processing. The slope can be interpreted as the rate by which information is accumulated or transmitted from the stimulus. It can also be understood in terms of the rate by which uncertainty is reduced over time through the gaining of information. It is measured in units of information per time (e.g., bits/s). In practical applications, researchers can compare the slopes of the iCRE(t) of several experimental conditions/tasks and make inferences on their relative processing efficiency (see for example Figure 5f). The condition/task with the steeper iCRE(t) slope is the one in which information gain (or entropy reduction) is more intense. This proposed measure of processing efficiency is consistent with extant definitions of capacity as a dynamic aspect of information processing that reflects the *intensity* or *energy* of a system [11,64].

## 8. Relations of the iCRE(t) to Other RT Functions

The iCRE(t) bears close affinity with two useful RT functions (see Luce’s classic book [18], p. 18). First, the residual entropy function has quantitatively the same value, though with an opposite sign, as the log survivor function:(15)$$\begin{array}{cc} & \;log[S(t)]\; \end{array}$$
which is practical for summarizing and plotting temporal aspects of RT data [18]. Second, the iCRE(t) is mathematically identical to the *integrated hazard function* for which the following equivalence has already been introduced:(16)$$\begin{array}{ccc} H(t) & = & -\log(S(t) \end{array}$$

The *instantaneous cumulative residual entropy function*
iCRE(t) and the *integrated hazard function* are gauged by the same mathematical equation and can be viewed as conveying a similar conceptual definition of processing efficiency or capacity. However, the two indices differ in their units of measurement. The former is measured in units of the to-be-processed elements, whereas the latter is measured in universal units of information (bits). This confers the iCRE(t) a marked advantage, because it allows researchers to compare performance across different conditions, tasks, or experimental procedures based on a common unit of measurement (e.g., bits). Previous applications of the hazard function to questions of human performance and capacity, in particular, have circumvented the problem of different units of measurement by using the ratio of (cumulative) hazard functions [50]. A ratio gives a pure quantity that gets rid of the units of measurement. Wenger and colleagues [50,51,65], for example, have harnessed Cox’s regression [66], which relies on the ratio of hazards, to model the psychological construct of processing capacity. Townsend and colleagues [12,45] have developed a *capacity coefficient* that circumvents the problem by expressing a ratio of cumulative hazard functions. With the iCRE(t) and CRE measures, comparisons across conditions, experiments, and tasks can be made directly.

Recently, Lappin and colleagues [24] have presented the *cumulative hazard function*
B(t). The function measures the cumulative work—detection progress—up to time *t*. They have also proposed an information theoretic interpretation of the derivative function *b(t)*:(17)$$\begin{array}{ccc} b(t) & = & \frac{f(t)}{S(t)})\times \frac{1}{log_{e}\left(2\right)} \end{array}$$
which measures the rate of change in the cumulative hazard function *B(t)* in units of bits/s. Note that the derivative of the cumulative hazard function is the hazard function itself, which, according to these authors, can be scaled to obtain units of bits. These functions bear close affinity to the measures developed here.

## 9. Simulations of the iCRE(t)

Figure 5a,f depicts the outcome of simulations with the exGaussian distribution, using the exact same parameter values used earlier. In these simulations, two hypothetical tasks/conditions are modeled that differ (in each simulation) with respect to one of the three exGaussian parameters (μ, σ, or τ). The results of the first simulation are presented in Figure 5a,b, where the two exGaussians differed only in the μ parameter, which implies a selective change in the means of the two distributions. Recall that in the exGaussian distribution, E(x)=μ+τ, whereas $Var\left(x\right)=\sigma^{2}+\tau^{2}$. The iCRE(t) of the fast task (plotted in pink) is shifted to the left relative to the slower task (plotted in blue), but the rate of information gain (reduction in entropy) of the two tasks, as indicated by the slopes of their iCRE(t) functions, is identical (0.0169 bans/ms).

The results of the second simulation are presented in Figure 5c,d, where the two exGaussians differed only in the value of their σ parameter. This manipulation selectively influences the variance of the distribution. The blue PDF distribution, with the higher standard deviation, yielded an iCRE(t) slope that was comparable to that observed in the pink distribution (0.0164 bans/ms).

The results of the third simulation are depicted in Figure 5e,f, where the two exGaussians differed only in the value of their τ parameter. This manipulation affects both the mean and the variance of the distribution. The blue distribution (with the larger τ value) exhibited a significantly slower rate of information gain (0.0084 bans/ms) than the pink distribution (0.0164 bans/ms).

The simulations exemplified the utility of our new entropy measure in uncovering important statistical effects that are not immediately discernible from the PDFs or the CRE functions. Notably, the rate of information gain can be the same across two experimental tasks/conditions even when one condition/task is slower than the other. This was demonstrated in the first simulation. Moreover, while higher variance in the PDF of one condition/task might entail larger values of total entropy (as indicated by the integral of the CRE in the second simulation), it does not necessarily imply a lower rate of information gain. Finally, we noted that when both mean and variance are increased, the information transmission rate can decrease. These observations are based on one type of distribution (the exGaussian), but they suggest intricate relations between the parameters of a distribution and its corresponding iCRE(t) function. These relations are not directly observable from the PDF or even CRE functions.

Lastly, an issue that deserves a comment concerns the wiggling of the iCRE(t) line at high values of *t*. As can be noted in Figure 5b,d,f, the line becomes jerky for long *t* values. Recall that this function is estimated directly from the survivor function S(t). It is likely that the number of data points at prolonged times is drastically reduced. Naturally, at longer times, there are only few instances that survive. This scarcity of data points can account for the relatively unstable estimation of the iCRE(t) function with these late *t* values.

## 10. Discussion

Human behavior is governed by unobserved internal structures (e.g., attention). These are hidden from direct observation, leaving only residual traces in behavior. In this context, one can appreciate the immense utility of response times (RT) in unearthing cognitive structures and in facilitating the development of quantitative models of human cognition and behavior. Most researchers use mean RT as the central dependent variable while ignoring the great deal of information lurking in the entire distribution. One notable property of entire RT distributions is their *entropy*, which can be of considerable theoretical importance. The concept of entropy has strong conceptual links to psychological notions such as mental energyand efficiency.However, to date, no valid measure of entropy on RT distributions has been developed. This is mainly because available elaborations of discrete entropy measures to differential entropy were fraught with problems and inconsistencies. Recent advances in information theory, and in particular the development of the *cumulative residual entropy* (CRE) [30], have allowed researchers in many areas to begin measuring differential entropy in continuous distributions [36,37]. The present study has taken the first step in applying the CRE to one of psychology’s most important dependent variables—response times (RT) distributions. Here, I have developed a set of novel measures of entropy on RT distributions and uncovered their close formal affinity with existing RT functions.

The entropy measures of RT distributions developed here have great theoretical and practical benefits in assessing the level of energy or effort invested by an observer in a given task. Potential contributions of such measures are numerous. For example, one can compare the amount of entropy exerted by two experimental conditions or by two populations. Another potential contribution is in relating measures of RT entropy to signatures of entropy in the brain [31]. In addition, the slope of the iCRE(t)—a measure of the rate by which information is gained or transmitted from a stimulus—can be particularly useful in estimating the energy needed to perform a task or in modeling internal mechanisms of processing. One area that can benefit from these developments is the study of parallel/serial systems and their overall capacity [67].

Many stochastic models of human judgment and choice take into consideration both response times and accuracy. A notable example is Ratcliff’s diffusion model [68]. Future work on the CRE in psychology should seek to incorporate the dependent variable of accuracy. A methodology for measuring information transmission with RT and accuracy does exist based on the notion of *speed accuracy tradeoff* (SAT) [64,69,70]—the observed regularity by which the probability of correct response increases as RTs get longer. The information *I* transmitted by the stimulus is computed as a function of accuracy P with the formula: $I=P\log_{2}P+\left(1-P\right)\log_{2}\left(1-P\right)$ at a given mean RT, and plotted against it. Typically, *I* increases monotonically with mean RT. The interpretation of this function is that the observer extracts more information as their responses are prolonged. This is an important measure, but it relies on mean RTs and not on the entire RT distribution. In addition, it is based on the premise that the speed–accuracy trade-off function is always positive—an assumption that may sometimes be violated [71]. The proposed CRE measures avoid these problems, because they are computed on entire distributions and make minimal assumptions about processing.

The present study has developed new measures of entropy on RT distributions, with the main focus being their interpretation as capacity indices. However, it should be highlighted that these entropy measures have a general purpose and can be potentially valuable in other applications where the notions of capacity or energy are not involved. For example, when fitting empirical RT distributions, the difference in entropy between observed and expected distributions can serve as a minimization function for the algorithm searching the parameter space. Another useful application is in the area of model comparison, where one can deploy differences in total entropy to decide between competing models. 

## Figures and Tables

**Figure 1 entropy-25-01239-f001:**
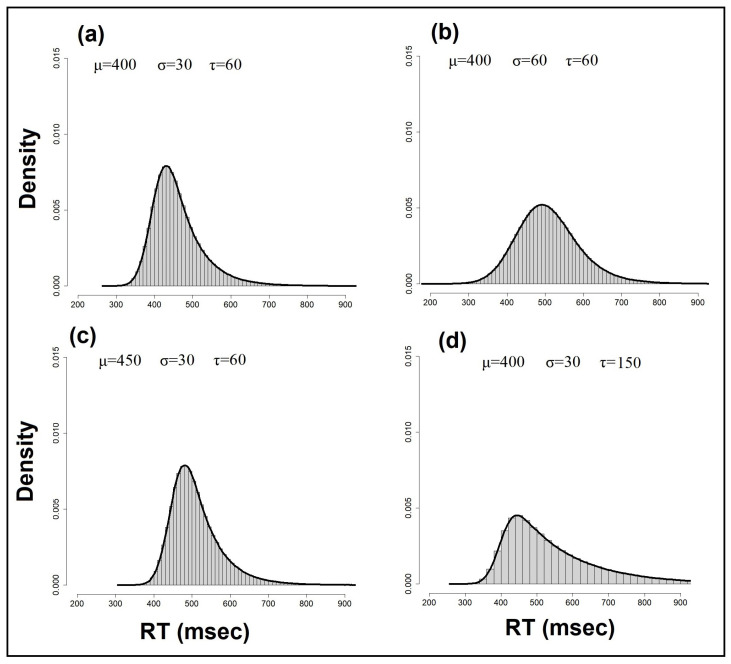
The exGaussian distribution is often chosen by psychologists to model RT distributions. The Gaussian part is represented by two parameters, the mean μ and standard deviation σ, and the exponential part by its mean τ. The mean and the variance of the exGaussian distribution are, respectively, E(x)=μ+τ and $Var\left(x\right)=\sigma^{2}+\tau^{2}$. The exGaussian provides an excellent fit to empirical data and captures the negatively skewed shape of RT distributions. The effects of its three parameters on the shape and location of the distribution are illustrated: (**a**) baseline parameters, (**b**) a change in the σ parameter, (**c**) a change in the location μ parameter, and (**d**) a change in the τ parameter.

**Figure 2 entropy-25-01239-f002:**
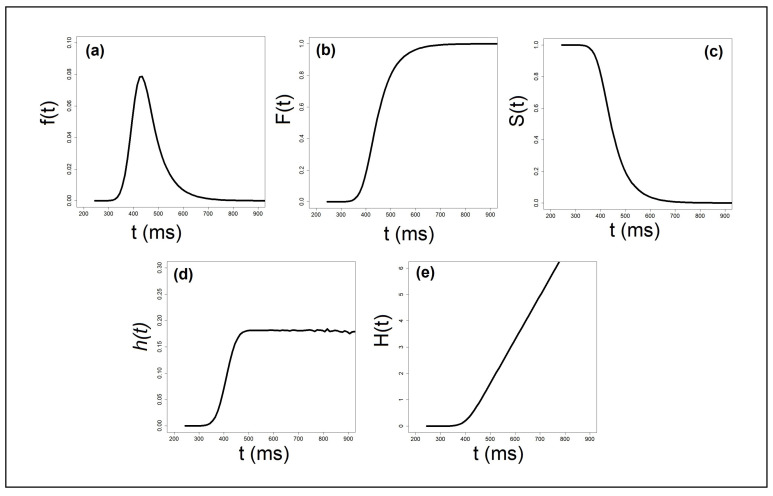
Five canonical RT functions simulated with the exGaussian distribution with parameters μ=400, σ=30 and τ=60: (**a**) probability density function (PDF), (**b**) cumulative density function (CDF), (**c**) survivor function, (**d**) hazard function, and (**e**) cumulative hazard function. See text for more details.

**Figure 3 entropy-25-01239-f003:**
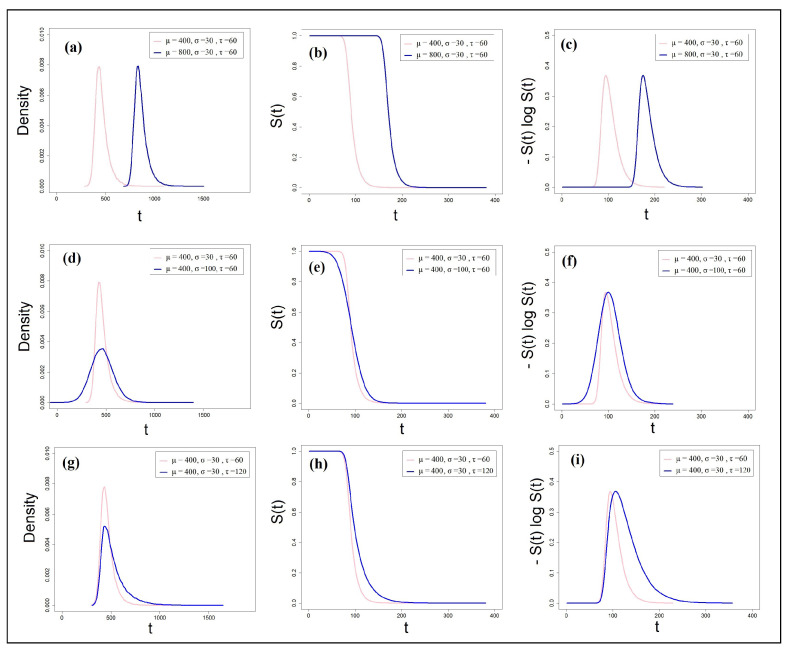
Simulations of PDFs, survivors, and CREs with two exGaussian distributions. Top panel: manipulation of the μ parameter: (**a**) probability density functions, (**b**) survivor functions, (**c**) CRE functions. Middle panel: manipulation of the σ parameter: (**d**) probability density functions, (**e**) survivor functions, (**f**) CRE functions. Bottom panel: manipulation of the τ parameter: (**g**) probability density functions, (**h**) survivor functions, (**i**) CRE functions.

**Figure 4 entropy-25-01239-f004:**
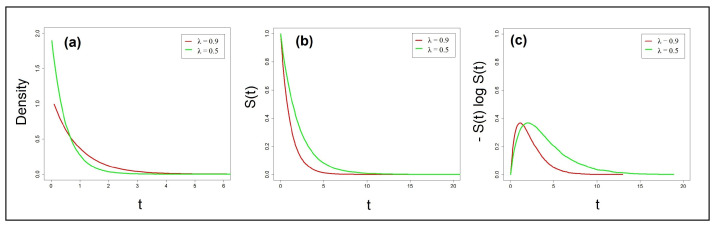
Simulations of PDFs, survivors, and CREs with two exponential distributions: (**a**) probability density functions, (**b**) survivor functions, and (**c**) their corresponding CRE functions.

**Figure 5 entropy-25-01239-f005:**
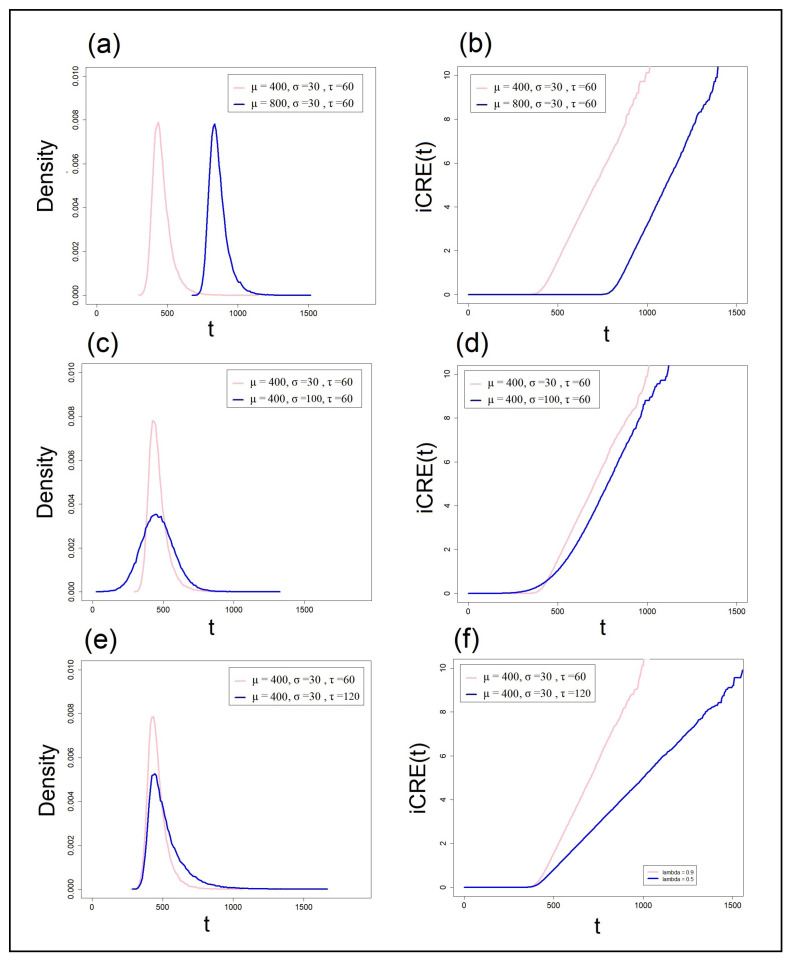
Simulations of the iCRE(t) function for the same exGaussian parameters as in Figure 3. (**a**) Two PDFs differing in μ, (**b**) iCRE(t) functions for the two pdfs differing in μ, (**c**) two PDFs differing in σ, (**d**) iCRE(t) functions for the two PDFs differing in σ, (**e**) two PDFs differing in τ, (**f**) iCRE(t) functions for the two pdfs differing in τ.

## Data Availability

Codes for generating the simulated data will be given upon request.

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
