# Peer review of "Quantifying Entropy in Response Times (RT) Distributions Using the Cumulative Residual Entropy (CRE) Function"

_entropy, 2023, doi:10.3390/e25081239_

Round 1
Reviewer 1 Report
Review of manuscript entropy-2535132, “Quantifying entropy in response times (RT) distributions using the cumulative residual entropy (CRE) function”, by Fitousi.
This is an important manuscript, and I endorse its publication. Minor revisions are advisable, but the central idea is insightful.
Because I have shared interest in promoting closely related ideas, and because I wish to cite this article for its focus on entropy and its explicit linkage between RT hazard functions and entropy, in the interest of full disclosure, I identify myself as the reviewer, Joe Lappin. Before reading this article, I was familiar with applications of hazard functions to RT distributions, was aware of their relevance to Shannon’s entropy-based theory of communication, and was aware of theoretical contrasts between this approach and standard theories of RT distributions. Even so, the author’s emphasis on entropy was stimulating and clarifying. The conceptual distinction between RT distributions as measures of process durations and as measures of response entropy (variability, uncertainty) is fundamentally important.
From my perspective, concern with the transformation from particular stochastic distributions (e.g., Weibull, lognormal, inverse Gaussian, ex-Gaussian, etc.) to the iCRE(t) functions is slightly confusing and potentially misleading. Similarly, the illustrations of relations between iCRE(t) and mean RT differences for ex-Gaussian distributions and time-parameter differences for exponential distributions are of little interest — because such simple effects are empirically unrealistic, and because one would not start with those distributions if the causal processes are thought to involve entropy rather than time duration. The present theory that RT distributions reflect entropy as a function of RT is, in my opinion, fundamentally different from the conception of RTs as measures of process durations as defined by the times from a specific stimulus to a particular response. They are mathematically different in several respects, and they entail different conceptions of the underlying causal mechanisms.
Some details to revise: The units of measure of iCRE(t) are not, as the author states, in units per time, but are in units defined at particular times. “Bans/ms” indicates a differential or derivative measure, but that is not what is given by iCRE(t) = –log [S(t)]. What are “bans”?
The statements on lines 312 – 318 on page 11, concerning an alleged conceptual distinction between iCRE(t) and H(t) make no sense to me. What is meant by the statement that units of the integrated hazard function are defined by “units of the to-be-processed items or inputs”? That is certainly not the case for Eq. 14. H(t) is a function of probability, not of some physical variable.
An empirical aside: We have found in all the many sets of empirical data we have examined that differentials or rates — increases in H(t) relative to increases in RT — measure meaningfully varying functions of RT that are lawfully influenced by experimental and task variables. The distinction between the cumulative information, H(t), and its derivative, h(t) or discrete approximations, is important both theoretically and empirically.
In several of the equations with integrals — e.g., Eqs. 4, 7, 8 — the variable of integration is shown as identical to the argument at which the function is defined. In Eq. 4, for example, H(t) is correctly defined as the limit of integration from 0 to t, but the functions h(t), f(t), and S(t) are integrated over a variable, say t’, not on a specific value of that variable. While this might look trivial, it is mathematically confusing not to distinguish the two things.
Minor typographical and grammatical errors are scattered throughout the manuscript. Proof reading is needed.
Author Response
Dear Professor Lappin,
I thank you for your positive evaluation of my work and your excellent suggestions and comments.
Please find attached a file with detailed responses to each of your comments.
Regards,
Danny

Reviewer 2 Report
Comments are attached in a pdf file.

Author Response
Dear Reviewer 2,
I thank you for your positive evaluation of my work and your excellent suggestions and comments.
Please find attached a file with detailed responses to each of your comments.
Regards,
Danny

Reviewer 3 Report
This interesting manuscript led me to realize that the adoption of a continuous representation of Shannon entropy, which is discrete, generates problems and that the Cumulative Residual Entropy (CRE) settles them. This manuscript shows that CRE goes much beyond settling the problems with the continuous representation of Shannon Entropy and affords an important statistical tool that psychologists can fruitfully use to study Reaction Times (RT).
The author makes an attractive illustration of the basic functions adopted to model RT distribution and I find very interesting his cumulative hazard function considering the effort necessary to realize a difficult task. I find very interesting his arguments on the instantaneous CRE and the adoption of his cumulative hazard function to address instantaneous CRE.
Minor remarks:
I find a misprint in Eq. (7) (missing the integration infinitesimal dx) and in Eq. (8) (missing the integration infinitesimal dt)
I warmly recommend this manuscript for publication.
Author Response
Dear Reviewer 3,
I thank you for your positive evaluation of my work and your excellent suggestions and comments.
Please find attached a file with detailed responses to each of your comments.
Regards,
Danny

Round 2
Reviewer 1 Report
This manuscript is certainly acceptable for publication. Further revisions are not required, in my opinion. My original judgment stands: This paper is an important contribution.
Reviewer 2 Report
The author has revised the manuscript according to the previous comments. I can now recommend acceptance.